# Efficiency of Using a Simulator for Ultrasound and Laser Dose Calculation in Physiotherapy

Francisco Javier Martin-Vega [1], Rocío Martín-Valero [2], Gloria Gonzalez-Medina [1,3,4,*], Inés Carmona-Barrientos [1], Cristina Garcia-Munoz [1] and Maria Jesus Vinolo-Gil [1,3,5]

1. Department of Nursing and Physiotherapy, University of Cadiz, 11009 Cadiz, Spain
2. Department of Physiotherapy, Faculty of Health Science, Ampliacion de Campus de Teatinos, University of Malaga, C/Arquitecto Francisco Peñalosa 3, 29071 Malaga, Spain
3. Research Unit, Biomedical Research and Innovation Institute of Cadiz (INiBICA), Puerta del Mar University Hospital, University of Cadiz, 11009 Cadiz, Spain
4. Research Group: CTS-986 Physical Therapy and Health (FISA), Institute of Research in Social Sustainable Development (INDESS), 11009 Cadiz, Spain
5. Rehabilitation Clinical Management Unit, Interlevels-Intercenters Hospital Puerta del Mar, Hospital Puerto Real, Cadiz Bay-La Janda Health District, 11006 Cadiz, Spain
* Correspondence: gloriagonzalez.medina@uca.es

**Abstract:** Background: In the context of the lockdowns due to COVID-19, e-learning has become the sole tool allowing learning objectives to be achieved successfully. However, for some subjects, training aided only by this type of tool encounters much difficulty, especially because of the experimental nature of such subjects. Aim: to assess the efficiency of a simulator for estimating dose calculation of ultrasound and laser based on surveys and a written test in a group of students. Methods: Surveys conducted voluntarily and anonymously by a group of students enrolled in the subject General Procedures in Physiotherapy I of the undergraduate degree in Physiotherapy. Furthermore, an objective test containing dose calculation problems for ultrasound and laser had to be solved. Prior to the completion of the objective test, the simulator for calculating ultrasound doses was provided to half of the subjects, whilst the other half were provided with the simulator for calculating laser doses, with both of which they were allowed to practice for a whole week. Results: Out of all the students enrolled in the first year of the undergraduate degree in Physiotherapy, a total of 38 students completed the surveys and 44 took part in the test for solving dose calculation problems. The surveys showed that a substantial number of students consider the use of a simulator for learning purposes efficient. This consideration was corroborated: the response times were reduced and the quantifications were the same for ultrasound, and better for laser. Conclusions: the use of a simulator for ultrasound and laser dose calculation is welcomed by a substantial number of students, and also represents a good additional tool when learning problem resolution.

**Keywords:** efficiency; simulator; ultrasound; laser; dose calculation; physiotherapy

## 1. Introduction

In 2007, David M. Gaba [1] defined simulation as "a technique, not a technology, to replace or augment real experiences by guided experiences evoking or replicating substantial aspects of the real world in a totally interactive way". Although today it is complicated to find standards allowing the analysis of the actual quality of simulators used in place of actual clinical practice [2], the use of learning applications in digital environments allows any student to autonomously practice with no space or time limits [3]. Previous experience reports satisfactory results, albeit with other types of simulators and specific pathologies [4–6]. Some of them even make it possible to assess the student's learning competences in much more specific ways than those currently used [7].

In the context of the lockdowns due to COVID-19, the e-learning has become the sole tool allowing learning objectives to be achieved successfully. However, for some subjects,

training aided only by this type of tool encounters much difficulty, especially because of the experimental nature of such subjects. This is the case of subjects such as electrotherapy, which is part of the undergraduate degree in Physiotherapy, in which determining doses of ultrasound or laser to be applied to a patient requires skilled practitioners in such processes in order to make the applied treatment effective. The students gain these skills by repetitively solving cases; an action which requires the appropriate "teacher-student" continuous interaction in terms of time and space, and which is not likely to take place under the current conditions. This is the reason a simulator has been created for estimating ultrasound and laser doses in order to assess its efficiency in the learning process of a group of students.

Therefore, the aim of this study is to assess the efficiency of a simulator for estimating dose calculation via ultrasound and laser based on surveys and a written test in a group of students.

## 2. Materials and Methods

### 2.1. Study Desing

This was a quasi-experimental longitudinal study with pretest and post-test design. It was conducted with a group of students from the School of Nursery and Physiotherapy of the University of Cadiz between February and April 2022, with the aim of assessing the use of the simulator as an efficient tool in the learning and practice process for ultrasound and laser doses calculation.

### 2.2. Selection Criteria

Inclusion criteria: Students enrolled in the subject General Procedures in Physiotherapy I of the school in which this study is conducted, who have voluntarily consented to participate. Students participated voluntarily.

Exclusion criteria: No access to a laptop or desktop computer, as well as a lack of the basic knowledge on how to work with Excel sheets.

### 2.3. Study Process

2.3.1. First Stage

After having shared the project with the students and explaining the topic regarding ultrasound and laser dose calculation and how to use the simulators, each student anonymously and individually completed an initial survey consisting of five items (Table 1), the objective of which is to obtain information on the efficiency they expect the simulator to perform within their learning process.

2.3.2. Second Stage

Once completed, all the subjects in the study will be divided into two groups. These two groups will be formed alphabetically based on their first surname. The first group will be individually provided with the simulator for ultrasound dose calculation, and the second with the simulator for laser dose calculation. Thus, the group using the simulator for calculating the ultrasound dose will have to simultaneously practice with no simulator, through traditional learning methods, the calculation of laser doses, whilst the group using the simulator for calculating the laser dose will need to practice following the traditional learning method for calculating the ultrasound dose. Afterwards, the students will have a week (seven days) to practice how to calculate ultrasound and laser dose aided by the simulator and the relevant traditional learning method, not having any contact with the professor during this period.

**Table 1.** Items in the initial survey. Source: Prepared by authors.

| Items | Question | Available Answers (Choose Only One) |
|---|---|---|
| Item 1 | Which changes do you think this new method will bring compared to the traditional method when learning how to solve dose calculation problems? | • None<br>• Few<br>• Some<br>• Many |
| Item 2 | Which level of efficiency do you think this new method provides? | • None<br>• Low<br>• Average<br>• High |
| Item 3 | Do you think this type of applications would be applied to other subjects? | • Yes<br>• No |
| Item 4 | How much time do you think you will have to spend in relation to the knowledge you are about to gain? | • Moderate<br>• Appropriate<br>• Excessive |
| Item 5 | How helpful do you think this application will be when solving dose calculation problems? | • Not at all<br>• Little<br>• Quite<br>• Much |

### 2.3.3. Third Stage

After this week, all the students will individually and anonymously complete a written test consisting of two problems on dose calculation, one for ultrasound, and another one for laser, with the aim of assessing their learning process. During the test, they will be asked to report the time needed to complete each problem. Finally, every student will individually and anonymously complete a final survey in order to obtain information on various aspects related to their learning experience through a simulator, as well as an assessment of the teacher related to the process of communicating information (Table 2). The flowchart below briefly and sequentially shows the process followed throughout the study (Figure 1).

**Table 2.** Items in the final survey. Source: Prepared by authors.

| Items | Question | Available Answers (Choose Only One) |
|---|---|---|
| Item 1 | Which changes do you think this new method has brought compared to the traditional method when learning how to solve dose calculation problems? | • None<br>• Few<br>• Some<br>• Many |
| Item 2 | Which level of efficiency do you think this new method has provided? | • None<br>• Low<br>• Average<br>• High |
| Item 3 | Do you think this type of applications would be applied to other subjects? | • Yes<br>• No |
| Item 4 | How much time has it taken to learn how to use this application in relation to the knowledge you have gained? | • Moderate<br>• Appropriate<br>• Excessive |
| Item 5 | How helpful do you think this application has been when solving dose calculation problems? | • Not at all<br>• Little<br>• Quite<br>• Much |
| Item 6 | In your opinion, how was the professor's communication when teaching how to solve dose calculation problems? | • Very poor<br>• Poor<br>• Average<br>• Good<br>• Very good |

**Table 2.** *Cont.*

| Items | Question | Available Answers (Choose Only One) |
|---|---|---|
| Item 7 | In your opinion, how was the professor's communication when teaching how to use the application for dose calculation? | • Very poor<br>• Poor<br>• Average<br>• Good<br>• Very good |

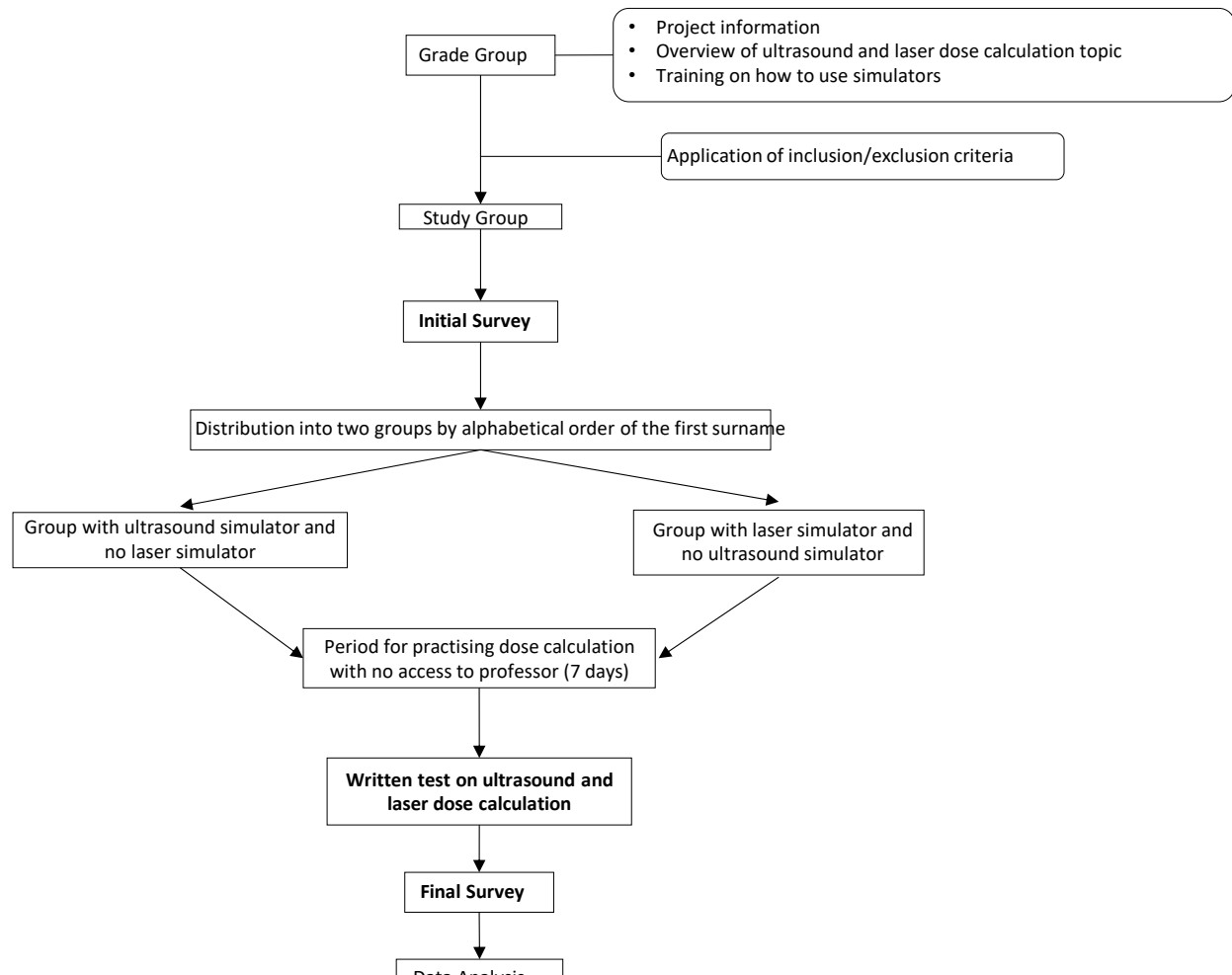

**Figure 1.** Flowchart depicting the study process. Source: Prepared by authors.

### 2.4. Data Analysis

The analysis consisted of a comparison of percentages of responses obtained between the initial and final survey, as well as a comparison of means recorded for the score and time required for completion of the written test on problem solving.

## 3. Results

### 3.1. Initial and Final Surveys Analysis

A total of 38 subjects took part in the initial survey and a total of 38 subjects took part in the final survey of this study.

Regarding Item 1, most of the students think that applying this simulator for learning how to calculate doses does contribute, with some changes compared to the traditional method. Although based on the comparison of the percentages from the initial and final surveys, these reflect that expected changes were higher than those experimented after

discovering and using the simulator (in the final survey, the ratio of students considering that the changes were few is higher than in the initial survey) (Table 3).

**Table 3.** Item 1: Changes this new method brings compared to the traditional method when learning how to solve dose calculation problems.

| Options | Initial Questionnaire. Number of Subjects (%) | Final Questionnaire. Number of Subjects (%) |
|---|---|---|
| None | 0 (0.00%) | 0 (0.00%) |
| Few | 3 (7.90%) | 8 (21%) |
| Some | 33 (86.84%) | 27 (71.00%) |
| Many | 2 (5.26%) | 3 (8.00%) |

Concerning Item 2, all the students questioned consider the level of efficiency of this new method to be average or high. In this case, a match was recorded between the level of efficiency they expected (initial survey) and the one experimented after its use (final survey) (Table 4).

**Table 4.** Item 2: Efficiency of this new method.

| Options | Initial Questionnaire. Number of Subjects (%) | Final Questionnaire. Number of Subjects (%) |
|---|---|---|
| None | 0 (0.00%) | 0 (0.00%) |
| Low | 0 (0.00%) | 0 (0.00%) |
| Average | 28 (73.68%) | 28 (73.68%) |
| High | 10 (26.32%) | 10 (26.32%) |

As far as Item 3 is concerned, the ratio of students willing to apply similar methods to those used in this study in other subjects is remarkably high. In the final survey, this ratio accounted for 100% (Table 5).

**Table 5.** Item 3: Willingness to use this type of application in other subjects.

| Options | Initial Questionnaire. Number of Subjects (%) | Final Questionnaire. Number of Subjects (%) |
|---|---|---|
| Yes | 36 (94.74%) | 38 (100%) |
| No | 2 (5.26%) | 0 (0.00%) |

Regarding Item 4, the majority of the students think the time necessary to learn how to calculate doses is appropriate. However, in the final survey there are more students advocating for moderate time spent instead of excessive, compared to the initial survey (Table 6).

**Table 6.** Item 4: Time to be spent compared to the knowledge gained.

| Options | Initial Questionnaire. Number of Subjects (%) | Final Questionnaire. Number of Subjects (%) |
|---|---|---|
| Moderate | 6 (15.79%) | 16 (42.10%) |
| Appropriate | 31 (81.58%) | 22 (57.90%) |
| Excessive | 1 (2.63%) | 0 (0.00%) |

With respect to Item 5, when solving calculation problems this method is seen as been quite helpful for a significant ratio of the students surveyed. Although in the final survey, an increase in the number of students considering it little or not at all helpful may be observed compared to the initial survey (Table 7).

**Table 7.** Item 5: How helpful this application is when solving dose calculation problems.

| Options | Initial Questionnaire. Number of Subjects (%) | Final Questionnaire. Number of Subjects (%) |
|---|---|---|
| Not at all | 0 (0.00%) | 1 (2.63%) |
| Little | 1 (3.70%) | 3 (7.90%) |
| Quite | 20 (74.07%) | 25 (65.79%) |
| Much | 6 (22.22%) | 9 (23.68%) |

When it came to assessing the professor's communication when explaining the solving calculation process and how the simulator works, the performance was rated as good or very good (Table 8). A similar rating was recorded when asking about the communication performed to explain how the simulator worked (Table 9).

**Table 8.** Item 6: Professor assessment when explaining how to solve dose calculation problems.

| Options | Final Questionnaire. Number of Subjects (%) |
|---|---|
| Poor | 0 (0.00%) |
| Average | 0 (0.00%) |
| Good | 11 (28.95%) |
| Very good | 27 (71.05%) |

**Table 9.** Item 7: Professor assessment when explaining how dose calculation simulator works.

| Options | Final Questionnaire. Number of Subjects (%) |
|---|---|
| Poor | 0 (0.00%) |
| Average | 0 (0.00%) |
| Good | 15 (39.47%) |
| Very good | 23 (60.53%) |

*3.2. Analysis of the Written Test*

The written test was taken by 44 subjects (22 of them evaluated the dose calculation simulator for ultrasound, and the remaining 22 the dose calculation simulator for laser). This test consisted of solving two calculation problems—one ultrasound—and a second—laser-related. Each problem included two sections. Each scored a maximum of 0.5 points. Therefore, the score for successfully solving each problem in the two sections was 1 point. Students had to note down the estimated time necessary to complete each problem.

When comparing the average marks and the average time spent completing the ultrasound problem with the laser problem, those with the ultrasound simulator obtained nearly the same score in the ultrasound problem (0.66 points) and the laser problem (0.64 points). However, they spent less time completing the ultrasound problem (7.10 min for those having the simulator, compared to 10.66 min for those not using it).

Nonetheless, those having the laser simulator received higher marks in the laser problem (0.84 points) than in the ultrasound problem (0.70 points), but they spent the same amount of time completing both problems (9.32 min).

When comparing the average marks and the time spent based on who had the simulator and who did not, in the case of the ultrasound problem, those having the ultrasound simulator received a similar mark (0.66 points) to those who did not have it (0.70 points), although they spent less time completing the problem (7.10 min for those having the simulator compared to 9.32 min for those who did not have it).

However, in the case of the laser problem, those having the laser simulator received higher marks and spent less time (0.84 points and 9.32 min) completing the problem compared to those who did not have it (0.64 points and 10.66 min).

## 4. Discussion

### 4.1. Initial and Final Surveys Analysis

When comparing the results recorded in the initial and final surveys concerning the contribution to the learning process of this new method compared to the traditional method, its efficiency and how helpful it is when solving calculation problems (Items 1, 2, and 5 respectively), it is observed that expected changes were higher than that experienced after discovering and using the simulator. Nevertheless, the results recorded concerning implementation of similar methods in other subjects, as well as spending a moderate amount of time to gain the necessary knowledge through this method (Items 3 and 4, respectively) confirm that students taking part in this study approve the use of this technique for learning purposes. The positive rating by the students to the communication process conducted by the teacher when describing how to solve dose calculation problems, as well as the use of the simulator, eliminate any potential bias arising from interferences in the process of conveying the necessary knowledge to participants. Even so, we must be careful when assessing the learning method used based on the results recorded in the surveys. As they have been completed online, they could be a little less reliable than if they had been completed face-to-face [8]. The results obtained in previous studies assumed the presence of students [7].

### 4.2. Initial and Final Surveys Analysis

Although completing surveys on-line may seem a reliable method, using other methods for obtaining information and complementing that given on the surveys could strengthen the validity of the answers given by the surveyed persons [9]. This is the reason for why all the students participating in the study were subject to a written test. Through solving problems and recording the time necessary to complete the test, the test seeks to analyze whether there are differences among those using the simulator and those following the traditional method considering that they did not have any chance to contact the professor during their study. The results recorded in the test depict an equivalent or higher scoring and less time taken to solve the problems among those students practicing with the simulator before taking the test.

Based on the results arising from this study, the concept that the online method is an efficient tool for learning purposes is shared by different authors [10–12] as far as new students are concerned [13]. For the sake of patient security, constraints do exist in real practice [5,6], however these are offset by the repetition of practical cases revisited digitally, which allow practitioners to make an in-depth analysis on how to implement the treatment and obtain feedback [14,15]. This could be also implemented by practitioners with the aim of facilitating their clinical daily job in situations going beyond the usual standards [16,17], since this simulator supports scheduling a treatment within a wide range of parameters allowing its use in non-standardized contexts, unlike other digital simulation procedures [18].

### 4.3. Strengths and Limitations

In our opinion, one of the key elements of this study's results lies in how the information on the topic was conveyed to the students, that is, face-to-face, and which was rated as "good" or "very good" in the final survey. Such rating eliminates any potential bias arising from interferences in the information and communication process to the students regarding the learning process for problems resolution, as well as using the simulator (verbal communication). Nevertheless, practicing with the simulator instead of having access to the professor (non-verbal communication) may have hindered students during the thinking process for solving calculation problems. This may be due to a potential deficiency in simulator's design [19,20]. As a consequence, and despite some studies supporting the prevalence of simulated training compared to traditional training [21], we believe the efficiency of this learning method is based on a combination of traditional elements, such as the teacher's face-to-face presentation of the topic [19], which have been complemented

by a more digitalized practice aided by the simulator [22]. Further, we consider that this combination may not replace real practice [23], but it may become a good introduction to it [24].

A prudent analysis of the results recorded is recommended, since the number of participants was extraordinarily small, as well as the year in which it was applied (the first year of the undergraduate degree). An increased number of subjects taking part in the study is recommended, if possible, from different years of the same degree, as well as the incorporation of several teachers into the information conveyance stage.

## 5. Conclusions

The survey conducted among the subjects taking part in this study describes that the use of the simulator for learning how to calculate ultrasound and laser doses is efficient and introduces some changes in the training process compared to the traditional method. It is regarded as quite helpful to solve dose calculation problems for which an adequate amount of time is taken in the learning and handling process. They show themselves willing to apply similar methods in other courses.

With regards to the test to which participants were subject, the use and handling of the simulator prior to taking the test led to an equivalent or higher score in the calculation problems being obtained according to the simulator they used, as well as a lower amount of time to solve such problems.

**Author Contributions:** Conceptualization, F.J.M.-V. and R.M.-V.; methodology, I.C.-B. and G.G.-M.; formal analysis, C.G.-M. and M.J.V.-G.; data curation, C.G.-M. and M.J.V.-G.; writing—original draft preparation, F.J.M.-V. and R.M.-V.; writing—review and editing, I.C.-B. and G.G.-M.; visualization, R.M.-V.; supervision, F.J.M.-V. All authors have read and agreed to the published version of the manuscript.

**Funding:** This research received no external funding.

**Informed Consent Statement:** Not applicable.

**Data Availability Statement:** Not applicable.

**Conflicts of Interest:** The authors declare no conflict of interest.

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
