# Peer review of "Efficiency of Using a Simulator for Ultrasound and Laser Dose Calculation in Physiotherapy"

_applsci, doi:10.3390/app12189096_

Round 1
Reviewer 1 Report
The authors present a good research paper.
- The relevance of the topic: Good.
- Introduction: Good.
- Methodology: Can be improved.
- Results: Good.
- Discussion: Good.
However, ACCEPT AFTER MINOR REVISION. In general, the paper follows an adequate structure and correct scientific support and can be published considering some limitations. The study is interesting in the field of new techniques for physiotherapy. However, there are a series of limitations that should be considered.
In the first place, carry out a review of the existing literature related to the subject, being essential to inquire into the MPDI – Sensors journal itself, since there are papers related to its manuscript that can help to improve it. Therefore, include those references, if any, especially from the last five years. Also, recommend reading some papers related to the topic of new techniques for physiotherapy.
Dixit, S., Goyal, B., Sharma, H., & Kholiya, K. (2022). A Review of Emerging Tools and Techniques in Physiotherapy. International Journal of Scientific Advances, 3(3), 412-415.
Fernández-Carnero, S., Martin-Saborido, C., Achalandabaso Ochoa-Ruiz de Mendoza, A., Ferragut-Garcias, A., Cuenca-Zaldivar, J. N., Leal-Quiñones, A., ... & Gallego-Izquierdo, T. (2021). The Role of Rehabilitative Ultrasound Imaging Technique in the Lumbopelvic Region as a Diagnosis and Treatment Tool in Physiotherapy: Systematic Review, Meta-Analysis and Meta-Regression. Journal of clinical medicine, 10(23), 5699.
Gaudreault, N., Lebel, K., Bédard, S., Daigle, F., Venne, G., & Balg, F. (2021). Using ultrasound imaging to assess novice physiotherapy students’ ability to locate musculoskeletal structures with palpation. Physiotherapy, 113, 53-60.
Willett, M., Duda, J., Fenton, S., Gautrey, C., Greig, C., & Rushton, A. (2019). Effectiveness of behaviour change techniques in physiotherapy interventions to promote physical activity adherence in lower limb osteoarthritis patients: a systematic review. PloS one, 14(7), e0219482.
Specific comments.
Title: It´s righ.
Abstract. Incorporate in the summary, a more precise sentence of the results.
In their 1. Introduction. This section presents the problem in a coherent and clear manner with the correct support of the scientific literature. However, it is convenient to update the references, since there are different works related to the subject and no mention is made, and it would even be interesting to mention the different existing studies related to of new techniques for physiotherapy and expand the introduction section. Also, it could be a future study of review.
In section 2. Methods. Modify the method section and incorporate the sections: Design.
- Study design. To write the design section, we recommend that you take some of the following methodologists as references.
Ato, M., López-García, J. J., & Benavente, A. (2013). A classification system for research designs in psychology. Anales de Psicología/Annals of Psychology, 29(3), 1038-1059.
Montero, I., & León, O.G. (2007). A guide for naming research studies in psychology. International Journal of Clinical and Health Psychology, 7(3), 847-862.
In their section 4. Results. Summary of study data and table are correct.
In section 5. Conclusion. Differentiate the discussion of the main conclusions of the work. To do this, you must create this section. And modify the limitations of the study and locate them in said section at the end. Also, they must be direct, and highlight the main contributions of the study.
In section 6. References. They should be reviewed and updated according to the publication standards.
With Kind Regards,
Reviewer 2 Report
The paper is moderat new in content, no have news for our domain, but i see that all requirements is in order
Reviewer 3 Report
A work of great originality.
English language and style are fine/minor spell check required
